# SELF: Language-Driven Self-Evolution for Large Language Models

## Abstract

Large Language Models (LLMs) have demonstrated remarkable versatility across various domains. To further advance LLMs, we propose 'SELF' (Self-Evolution with Language Feedback), a novel approach that enables LLMs to self-improve through self-reflection, akin to human learning processes. SELF initiates with a meta-skill learning process that equips the LLMs with capabilities for self-feedback and self-refinement. Subsequently, the model undergoes an iterative process of self-evolution. In each iteration, it utilizes an unlabeled dataset of instructions to generate initial responses. These responses are enhanced through self-feedback and self-refinement. The model is then fine-tuned using this enhanced data. The model undergoes progressive improvement through this iterative self-evolution process. Moreover, the SELF framework enables the model to apply self-refinement during inference, which further improves response quality. Our experiments in mathematics and general tasks demonstrate that SELF can enhance the capabilities of LLMs without human intervention. The SELF framework indicates a promising direction for the autonomous evolution of LLMs, transitioning them from passive information receivers to active participants in their development.

## 1 Introduction

Large Language Models (LLMs), like ChatGPT (OpenAI, 2022) and GPT-4 (OpenAI, 2023), stand at the forefront of the AI revolution, transforming our understanding of machine-human textual interactions and redefining numerous applications across diverse tasks. Despite their evident capabilities, achieving optimum performance remains a challenge.

The intrinsic learning mechanisms employed by humans inspire optimal LLM development. A self-driven learning loop is inherent in humans when confronted with new challenges, involving initial attempts, introspection-derived feedback, and refinement of behavior as a result. In light of this intricate human learning cycle, one vital question arises: "Can LLMs emulate human learning by harnessing the power of self-refinement to evolve their intrinsic abilities?" Fascinatingly, a recent study (Ye et al., 2023) in top-tier LLMs such as GPT-4 has revealed emergent meta-skills for self-refinement, signaling a promising future direction for the self-evolution of LLMs. Despite this, current methods for LLM development typically rely on a single round of instruction fine-tuning (Wei et al., 2021; Zhou et al., 2023) with meticulously human-crafted datasets and reinforcement learning-based methods (Ouyang et al., 2022) that depend on an external reward model. These strategies not only require extensive resources and ongoing human intervention but also treat LLMs as mere passive repositories of information. These limitations prevent these models from realizing their intrinsic potential and evolving toward a genuinely autonomous, self-sustaining evolutionary state.

Our goal is to reveal the potential of LLMs for autonomous self-evolution by introducing a self-evolving learning framework called "SELF" (Self-Evolution with Language Feedback). Fig. 1 illustrates how SELF emulates the self-driven learning process with introspection and self-refinement. Through self-feedback and self-refinement, LLMs undergo iterative self-evolution as they learn from the data they synthesize. Furthermore, SELF employs natural language feedback to improve the model's responses during inference. This innovative framework can enhance models' capabilities without relying on external reward models or human intervention. Self-feedback and self-refinement

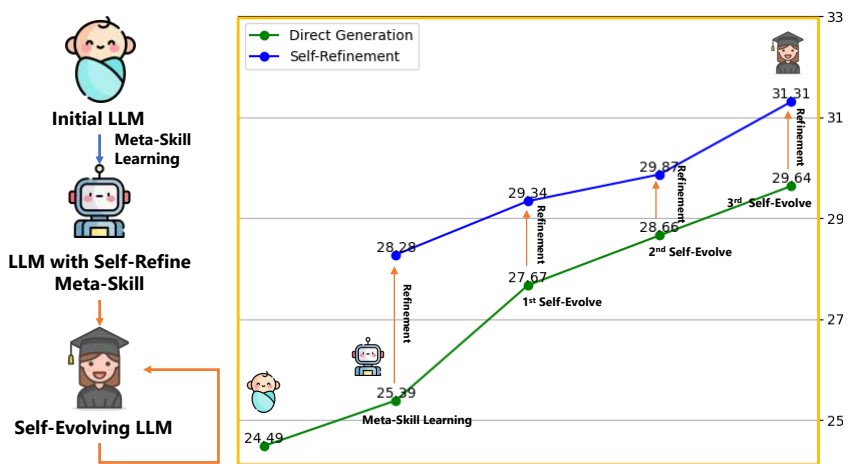

Figure 1: Evolutionary Journey of SELF: An initial LLM progressively evolve to a more advanced LLM equipped with a self-refinement meta-skill. By continual iterations (1st, 2nd, 3rd) of self-evolution, the LLM progresses in capability (24.49% to 31.31%) on GSM8K.

are integral components of the SELF framework. Equipped with these meta-skills, the model undergoes progressive self-evolution through iterative training with self-curated data. Evolution training data is collected by the model's iterative response generation and refinement processes. A perpetually expanding repository of self-curated data allows the model to enhance its abilities continuously. Data quality and quantity are continually improved, enhancing the intrinsic capabilities of LLMs. These meta-skills enable LLMs to enhance response quality through self-refinement during inference. As a result of the SELF framework, LLMs are transformed from passive data recipients into active participants in their evolution. The SELF framework not only alleviates the necessity for labor-intensive manual adjustments but also fosters the continuous self-evolution of LLMs, paving the way for a more autonomous and efficient training paradigm.

We evaluate SELF in mathematical and general domains. In the mathematical domain, SELF notably improved the test accuracy on GSM8k (Cobbe et al., 2021) from 24.49% to 31.31% and on SVAMP (Patel et al., 2021) from 44.90% to 49.80%. In the general domain, SELF increased the win rate on Vicuna testset (Lianmin et al., 2023) from 65.0% to 75.0% and on Evol-Instruct testset (Xu et al., 2023) from 48.6% to 55.5%. There are several insights gained from our experiments. First, SELF can continuously enhance the performance of models in generating direct responses through iterative self-evolution training. Second, meta-skill learning is essential for the model to acquire the ability for self-feedback and self-refinement. By self-refinement during inference, the model can consistently improve its response. Finally, meta-skill learning enhances the model's performance in generating direct responses. The model's generalization can be improved by providing language feedback to correct its mistakes.

The following key points summarize our contributions: (1) SELF is a framework that empowers LLMs with self-evolving capabilities, allowing for autonomous model evolution without human intervention. (2) SELF facilitates self-refinement in smaller LLMs, even with challenging math problems. The capability of self-refinement was previously considered an emergent characteristic of top-tier LLMs. (3) We demonstrate SELF's superiority, progressively demonstrating its ability to evolve intrinsic abilities on representative benchmarks and self-refinement capability.

## 2 RELATED WORKS

**Self-consistency** Self-consistency (Wang et al., 2022a) is a straightforward and effective method to improve LLMs for reasoning tasks. After sampling a variety of reasoning paths, the most consistent answer is selected. Self-consistency leverages the intuition that a complex reasoning problem typically admits multiple ways of thinking, leading to its unique correct answer. During decoding, self-consistency is closely tied to the self-refinement capability of LLMs, on which our method is based. Unlike self-consistency, self-refinement applies to a broader range of tasks, going beyond reasoning tasks with unique correct answers.

**Online Self-improvement for LLMs**   Various research efforts have been undertaken to enhance the output quality of LLMs through *online self-improvement* (Shinn et al., 2023; Madaan et al., 2023; Ye et al., 2023; Chen et al., 2023; Ling et al., 2023). The main idea is to generate an initial output with an LLM. Then, the same LLM provides feedback on its output and employs this feedback to refine its initial output. This process can be iterative until the response quality is satisfied.

While simple and effective, *online self-improvement* necessitates multi-turn inference for refinement, leading to increased computational overhead. Most importantly, *online self-improvement* does not prevent the model from repeating previously encountered errors, as the model's parameters remain unchanged. In contrast, SELF is designed to enable the model to learn from its self-improvement experiences.

**Human Preference Alignment for LLMs**   The concept of "Alignment", introduced by (Leike et al., 2018), is to train agents to act in line with human intentions. Several research efforts (Ouyang et al., 2022; Bai et al., 2022; Scheurer et al., 2023) leverage Reinforcement Learning from Human Feedback (RLHF) (Christiano et al., 2017). RLHF begins with fitting a reward model to approximate human preferences. Subsequently, an LLM is finetuned through reinforcement learning to maximize the estimated human preference of the reward model. RLHF is a complex procedure that can be unstable. It often requires extensive hyperparameter tuning and heavily relies on humans for preference annotation. Reward Ranked Fine-tuning (RAFT) utilizes a reward model to rank responses sampled from an LLM. Subsequently, it fine-tunes the LLM using highly-ranked responses (Dong et al., 2023). However, scalar rewards provide limited insights into the detailed errors and optimization directions, which is incredibly impractical for evaluating complex reasoning tasks involving multiple reasoning steps (Lightman et al., 2023). Instead, in this work, we propose to leverage natural language feedback to guide LLMs for self-evolution effectively.

**Reinforcement Learning Without Human Feedback in LLMs**   Recent advancements in LLMs have explored Reinforcement Learning (RL) approaches that do not rely on human feedback. LLMs are employed to assess and score the text they generate, which serves as a reward in the RL process (Pang et al., 2023). LLMs are updated progressively through online RL in interacting with the environment in Carta et al. (2023). The connection between conventional RL research and RLHF in LLMs is discussed by Sun (2023). While RL methods also enable automatic learning, they may not capture the nuanced understanding and adaptability offered by natural language feedback, a key component of SELF.

## 3   METHOD

As depicted in Fig. 1 and Fig. 2, the SELF framework aligns the model and enhances its inherent capabilities through a two-stage learning phase: (1) **Meta-skill Learning Phase**: This phase equips the model with essential meta-skills for self-feedback and self-refinement, laying a foundation for self-evolution. (2) **Self-Evolution Phase**: With the acquired meta-skills, the model progressively improves through multiple iterations of the self-evolution process. Each iteration begins with the model autonomously creating high-quality training data. Then, the model is fine-tuned using this data. The process is further illustrated in Alg. 1 in Appendix A.4.

### 3.1   META-SKILL LEARNING

The meta-skill learning stage aims to instill two essential meta-skills into LLMs:

(1) **Self-Feedback Ability**: This skill enables LLMs to evaluate their responses critically, laying the foundation for subsequent refinements. Self-feedback also enables the model to evaluate and filter out low-quality self-evolution training data (§ 3.2.1).

(2) **Self-Refinement Ability**: Self-refinement involves the model optimizing its responses based on self-feedback. This ability has two applications: (1) improving model performance by refining the models' outputs during inference (§ 3.2.3) and (2) enhancing the quality of the self-evolution training corpus (§ 3.2.1).

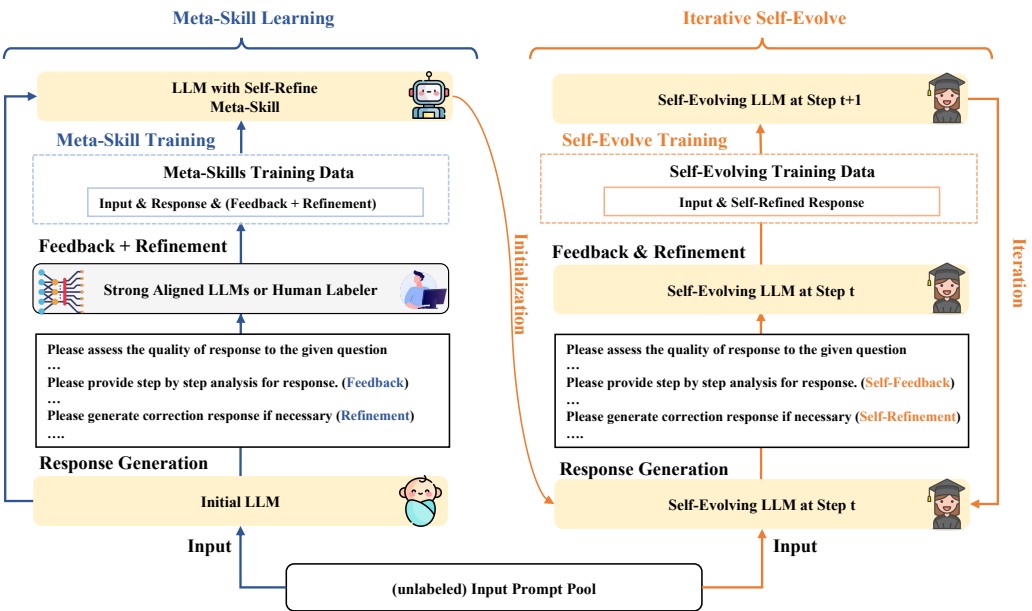

Figure 2: Illustration of SELF. The "Meta-Skill Learning" (left) phase empowers the LLM to acquire meta-skills in self-feedback and self-refinement. The "Self-Evolution" phase (right) utilizes meta-skills for self-evolution training with self-curated data, enabling continuous model enhancement.

These meta-skills are acquired by fine-tuning the model using the **Meta-Skill Training Corpus**. Details are provided in § 3.1.1. The resulting model is denoted as $M_{meta}$. Meta-skill learning establishes a foundation for the model to initiate subsequent self-evolution processes.

### 3.1.1 META-SKILL TRAINING CORPUS

The construction of the meta-skill learning corpus $D_{meta}$ involves the following elements: (1) An initial unlabeled prompt corpus $D_{unlabeled}$; (2) An initial LLM denoted as $M_{initial}$; (3) A strong LLM or human labeler $L$ tasked with evaluating and refining the responses of $M_{initial}$.

Specifically, the construction process operated in the following steps: (1) For each unlabeled prompt $p$ in $D_{unlabeled}$, the initial model $M_{initial}$ generates a initial response $r$. (2) The annotator $L$ provides evaluation feedback $f$ for the initial response $r$, then produces a refined answer $\hat{r}$ according to the feedback $f$. (3) Each instance in the meta-skill training data corpus $D_{meta}$ takes the form $(p, r, f, \hat{r})$, representing the process of response evaluation and refinement. An example instance of $D_{meta}$ is provided in Appendix A.3.

The data structure in $D_{meta}$ differs from the standard question-answering format, potentially weakening the model's ability to provide direct responses. We add a pseudo-labeled QA dataset denoted as $D_{QA}$ to alleviate this issue. This dataset consists of pairs of questions $p$ and refined answers $\hat{r}$. Notably, $D_{QA}$ is derived from the LLM-labeled $D_{meta}$ and does not include any human-annotated ground-truth data. This data integration strategy ensures a balanced emphasis on direct generation and self-refinement capability.

We prompt the LLM labeler $L$ with the following template to generate feedback and refinement [1]:

---

[1]This prompt is designed for the math domain. Please refer to A.5 for the prompt of the general domain.

> **Prompt for feedback and refinement:**
> **(Feedback)** Please assess the quality of the response to the given question.
> Here is the question: $p$.
> Here is the response: $r$.
> Firstly, provide a step-by-step analysis and verification for response starting with "Response Analysis:".
> Next, judge whether the response correctly answers the question in the format of "judgment: correct/incorrect".
> **(Refinement)** If the answer is correct, output it. Otherwise, output a refined answer based on the given response and your assessment.

## 3.2 Self-Evolution Process

The model $M_{meta}$, equipped with meta-skills, undergoes progressive improvement through multiple iterations of the self-evolution process. Each iteration of the self-evolution process initiates with the model autonomously creating high-quality training data (§ 3.2.1). With an unlabeled dataset of prompts, the model generates initial responses and then refines them through self-feedback and self-refinement. These refined responses, superior in quality, are then utilized as the training data for the model's subsequent self-evolution training (§ 3.2.2).

### 3.2.1 Self-Evolution Training Data

A corpus of unlabeled prompts is needed for self-evolution training. Given that real-world prompts are often limited, we employ Self-Instruct (Wang et al., 2022b) to generate additional unlabeled prompts. We denote $M_{self}^t$ as the model at the $t$-th iteration. In the first iteration of self-evolution, we initialize $M_{self}^0$ with $M_{meta}$. For each unlabeled prompt, the model $M_{self}^t$ generates a response, which is subsequently refined through its self-refinement ability to produce the final output $\hat{r}_{self}$. The prompt and self-refined response pairs, denoted as $(p_{self}, \hat{r}_{self})$, are subsequently incorporated into the self-evolution training dataset $D_{self}^t$ for subsequent self-evolution processes.

**Data Filtering with Self-feedback:** To enhance the quality of $D_{self}^t$, we leverage the self-feedback capability of $M_{self}^t$ to filter out low-quality data. Specifically, $M_{self}^t$ applies self-feedback to the self-refined data $\hat{r}_{self}$, and only those responses evaluated as qualified are retained.

After each iteration of self-evolution training, the model $M_{self}$ undergoes capability improvements. This leads to the creation of a higher-quality training corpus for subsequent iterations. Importantly, this autonomous data construction process obviates the need for more advanced LLMs or human annotators, significantly reducing manual labor and computational demands.

### 3.2.2 Self-Evolution Training Process

At each iteration $t$, the model undergoes self-evolution training with the updated self-curated data, improving its performance and aligning it more closely with human values. Specifically, we experimented with two strategies for self-evolution training:

(1) *Restart Training*: In this approach, we integrate the meta-skill learning data $D_{meta}$ and the accumulated self-curated data from all previous iterations — denoted as $\{D_{self}^0, D_{self}^1, ..., D_{self}^t\}$ to initiate the training afresh from $M_{initial}$.

(2) *Continual Training*: Here, utilizing the newly self-curated data $D_{self}^t$, we continue the training of the model from the preceding iteration, represented as $M_{self}^{t-1}$. We also incorporate $D_{meta}$ into continual training to mitigate the potential catastrophic forgetting of meta-skills.

The impact of these two divergent training strategies is thoroughly analyzed in our experiments in Appendix A.8.

### 3.2.3 Response Refinement during Inference

Equipped with the meta-skills for self-feedback and self-refinement, the model can conduct self-refinement during inference. Specifically, the model generates an initial response and then refines it using self-refinement, akin to the method described in § 3.1. Response refinement during inference consistently improves the model's performance as shown in § 4.2.

## 4 Experiments

We begin with an introduction to the experimental settings (§ 4.1), encompassing the evaluation data, baseline model, and model variations. In § 4.2, we present our main experiment to show the efficacy of SELF. § 4.3 demonstrates the incremental performance enhancements observed throughout self-evolution processes.

Given space limitations, we conduct several experiments to verify the SELF framework and include their details in the Appendix. We verify the effect of different meta-skill training corpus construction methods in Appendix A.6. Appendix A.7 shows the impact of filtering strategies when constructing the self-evolution corpus. Appendix A.8 evaluates the impact of divergent self-evolution training strategies as described in § 3.2.2. We demonstrate that SELF outperforms supervised fine-tuning in Appendix A.9. We explore how SELF performs with different starting model qualities in Appendix A.10 to exhibit the scalability of the SELF framework. In Appendix A.11, we investigate how the quality of the meta-skill learning corpus influences self-evolution training. We compare the effect of training with a single round of self-evolution versus training iteratively in Appendix A.12.

### 4.1 Experiment Settings

#### 4.1.1 Evaluation Benchmarks

We focus on two representative mathematical benchmarks and two general benchmarks: **GSM8K** (Cobbe et al., 2021) contains high-quality, linguistically diverse grade school math word problems crafted by expert human writers, which incorporates approximately 7.5K training problems and 1K test problems. The performance is measured by accuracy (%). **SVAMP** (Patel et al., 2021) is a challenge set for elementary Math Word Problems (MWP). It is composed of 1000 test samples. The evaluation metric is accuracy (%). **Vicuna testset** (Lianmin et al., 2023) is a benchmark for assessing instruction-following models, containing 80 examples across nine skills in mathematics, reasoning, and coding. **Evol-Instruct testset** (Xu et al., 2023) includes 218 real-world human instructions from various sources, offering greater size and complexity than the Vicuna testset.

#### 4.1.2 Setup and Baselines

The **complete SELF framework** includes meta-skill training with $D_{meta}$, three iterations of self-evolution training, and optional self-refinement during inference. Our evaluation primarily focuses on assessing how self-evolution training can progressively enhance the capabilities of the underlying LLMs. We note that the SELF framework is compatible with all LLMs. In this study, we perform the experiment with **Vicuna-7b** (Chiang et al., 2023) , which stands out as one of the most versatile open instruction-following models. **Vicuna-7b**, fine-tuned from LLaMA-7b (Touvron et al., 2023), will be referred to simply as 'Vicuna' in subsequent sections. One of our baseline model is **Vicuna + $D_{QA}$** which are **Vicuna-7b** fine-tuned with the pseudo-labeled question-answer data $D_{QA}$ (§ 3.1.1). We also compare SELF with the Self-Consistency (Wang et al., 2022a) approach. We note that all model training utilized the same training hyperparameters shown in Appendix A.1.1.

For building the meta-skill training corpus $D_{meta}$, we utilize GPT-4 due to its proven proficiency in refining responses (An et al., 2023). Please refer to Appendix A.1.2 for more details about $D_{QA}$ and unlabeled prompts utilized in self-evolution training.

Additionally, we compare SELF with RLHF. We utilize the RLHF implementation from trlx[2]. We apply the same SFT model, **Vicuna + $D_{QA}$** as described above, for both SELF and RLHF. The re-

---

[2]https://github.com/CarperAI/trlx

ward model is initialized from **Vicuna-7b** and is fine-tuned using pair-wise comparison data derived from the meta-skill training corpus $D_{meta}$ (§ 3.1.1), where the refined response $\hat{r}$ is presumed to be better than the original one $r$.

## 4.2 MAIN RESULT

### 4.2.1 MATH TEST

Table 1: Experiment results on GSM8K and SVAMP comparing SELF with other baseline methods. Vicuna + $D_{QA}$ means Vicuna fine-tuned on $D_{QA}$.

| Model | Self-Evolution | Self-Consistency | Self-Refinement | GSM8K(%) | SVAMP(%) |
|---|---|---|---|---|---|
| Vicuna | | | | 16.43 | 36.40 |
| | | ✓ | | 19.56 | 40.20 |
| | | | ✓ | 15.63 | 36.80 |
| Vicuna + $D_{QA}$ | | | | 24.49 | 44.90 |
| | | ✓ | | 25.70 | 46.00 |
| | | | ✓ | 24.44 | 45.30 |
| Vicuna + $D_{QA}$ + SELF (Ours) | ✓ | | | 29.64 | 49.40 |
| | ✓ | ✓ | | 29.87 | 50.20 |
| | ✓ | | ✓ | 31.31 | 49.80 |
| | ✓ | ✓ | ✓ | **32.22** | **51.20** |

In Table 1, we present an experimental comparison of SELF against baseline models, as detailed in Section 4.1.2. This comparison elucidates SELF's effectiveness in enhancing LLM performance through self-evolution and offers several key insights:

**(1) Self-Evolution Enhances LLM:** Vicuna + $D_{QA}$ + SELF significantly outperforms its baseline Vicuna + $D_{QA}$ (24.49% $\xrightarrow{+5.15\%}$ 29.64% on GSM8K and 44.90% $\xrightarrow{+4.5\%}$ 49.40% on SVAMP), showcasing self-evolution's potential in LLMs' optimization.

**(2) SELF Instills Meta-Capability in LLMs:** The integration of self-refinement into Vicuna + $D_{QA}$ + SELF results in a notable performance boost (29.64% $\xrightarrow{+1.67\%}$ 31.31%), while baseline models show minimal or negative changes via self-refinement. We also provide a case analysis for the limited self-refinement ability in baseline models in Appendix A.2. This indicates that SELF instills advanced self-refinement capabilities into smaller models like Vicuna (7B), previously limited to larger LLMs (Ye et al., 2023) like GPT-4.

**(3) Pseudo-Labeled $D_{QA}$ Enhances Performance:** The inclusion of pseudo-labeled QA data $D_{QA}$ enhances Vicuna's performance, suggesting that pseudo-labeled QA data help in learning task-specific information.

**(4) SELF can work with Self-Consistency:** SELF works effectively with self-consistency, improving accuracy across models. The base Vicuna model, which may have uncertainties in its outputs, shows notable improvement with self-consistency, achieving a +3.13% increase. As the model progresses through self-evolution training and becomes more capable of generating correct math answers, the benefit from self-consistency diminishes. Combining self-refinement with self-consistency further elevates performance (e.g., 29.64% $\xrightarrow{+2.58\%}$ 32.22% on GSM8K), indicating that these two strategies can complement each other effectively.

### 4.2.2 COMPARISON WITH RLHF

In Table 2, we compare the performance of SELF with RLHF. We note that the SELF result in Table 2 differs from those in Table 1. This discrepancy arises because the experiments in Table 2 utilized data solely from the initial round of self-evolution training. As Table 2 shows, RLHF achieves a 25.55% accuracy on GSM8K, which is lower than the 27.67% performed by SELF. We observe that the reward model often fails to identify the correctness of the response, which limits performance

Table 2: Comparison of SELF and RLHF on GSM8K

| Method | Acc. of Feedback (%) | Acc. on GSM8K(%) |
|---|---|---|
| SFT (Vicuna + $D_{QA}$) | - | 24.49 |
| RLHF | 24 | 25.55 |
| SELF | **72** | **27.67** |

improvements. On the GSM8K test set, for incorrect answers produced by the SFT model (Vicuna + $D_{QA}$), the reward model only identifies 24% of them as incorrect, i.e., the reward model assigns lower scalar rewards to incorrect answers compared to correct answers. In contrast, SELF utilizes informative natural language feedback to provide a more accurate assessment. It correctly identifies 72% of incorrect answers.

### 4.2.3 GENERAL TEST

We expanded the evaluation of the SELF framework to include general domain benchmarks, explicitly using the Vicuna and Evol-Instruct test sets. Three configurations of the Vicuna model are evaluated: Vicuna, Vicuna + $D_{QA}$, and Vicuna + $D_{QA}$ + SELF. We utilized GPT-4 to evaluate the models' responses on both test sets. We follow the assessment methodology proposed by (Xu et al., 2023), which mitigated the order bias present in the evaluation procedures described in (Chiang et al., 2023).

The results are depicted in Figure 3. In this figure, blue represents the number of test cases where the model being evaluated is preferred over the baseline model (Vicuna), as assessed by GPT-4. Yellow denotes test cases where both models perform equally, and pink indicates the number of test cases where the baseline model is favored over the model being evaluated.

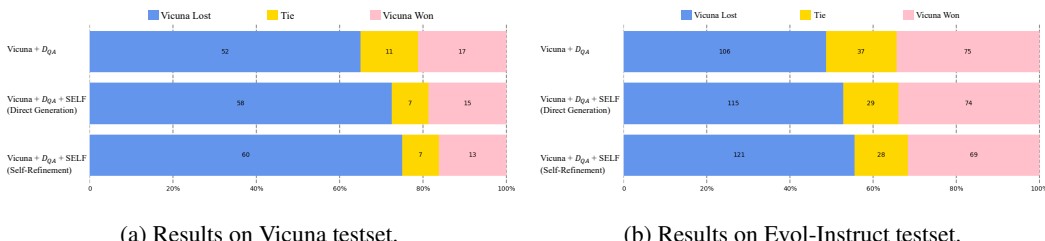

(a) Results on Vicuna testset.

(b) Results on Evol-Instruct testset.

Figure 3: Results on Vicuna testset and Evol-Instruct testset

In the Vicuna testset, Vicuna + $D_{QA}$ improved its win/tie/loss record from 52/11/17 to 58/7/15 with the addition of SELF. This translates to a win rate increase from 65.0% to 72.5%. After self-refinement, the record improved to 60/7/13, corresponding to a win rate of 75.0%. In the Evol-Instruct testset, Vicuna + $D_{QA}$ initially had a win/tie/loss record of 106/37/75, a win rate of about 48.6%. With SELF, this improved to 115/29/74, increasing the win rate to approximately 52.8%. Applying self-refinement, the record improved further to 121/28/69, equating to a win rate of 55.5%.

These findings in general domains highlight the SELF framework's adaptability and robustness, particularly when self-refinement is employed, showcasing its efficacy across varied test domains.

### 4.3 ABLATION STUDY FOR SELF

The SELF framework endows LLMs with an inherent capability through a structured, two-phase learning process. We conduct ablation experiments on SVAMP and GSM8K datasets to assess the incremental benefits of each stage. As depicted in Table 3, the framework facilitates gradual performance improvements through successive SELF stages. A checkmark ✓ in a column denotes the additive adoption of the corresponding setting in that training scenario. Observations are highlighted below:

Table 3: Performance comparisons of SELF under various training scenarios. Arrows indicate the improvement from direct generation to self-refinement: "direct generation $\rightarrow$ self-refinement"

| SVAMP (%) | GSM8K (%) | Meta-Skill Learning | | Self Evolution Process | | |
|---|---|---|---|---|---|---|
| | | $D_{QA}$ | $D_{meta}$ | 1st round | 2nd round | 3rd round |
| 36.4 | 16.43 | | | | | |
| 44.9 | 24.49 | ✓ | | | | |
| $46.8 \rightarrow 47.0$ | $25.39 \rightarrow 28.28$ | ✓ | ✓ | | | |
| $47.8 \rightarrow 48.0$ | $27.67 \rightarrow 29.34$ | ✓ | ✓ | ✓ | | |
| $48.9 \rightarrow 49.0$ | $28.66 \rightarrow 29.87$ | ✓ | ✓ | ✓ | ✓ | |
| $\mathbf{49.4 \rightarrow 50.2}$ | $\mathbf{29.64 \rightarrow 31.31}$ | ✓ | ✓ | ✓ | ✓ | ✓ |

**(1) Integration of Meta-skill Training Data $D_{meta}$ Elevates Direct QA:** Incorporating data detailing the feedback-refinement process ($D_{meta}$) in meta-skill training notably enhances direct response quality (+1.9% on GSM8K and +2.28% on SVAMP) in comparison to using $D_{QA}$ alone. This underscores the interesting finding that arming the model with self-refinement meta-capability implicitly elevates its capacity to discern the standard of a good answer and generate superior responses, even without explicit self-refinement.

**(2) Continuous Improvement through Self-Evolution:** The results reveal that three self-evolution rounds consecutively yield performance enhancements (e.g., $25.39\% \xrightarrow{+2.28\%} 27.67\% \xrightarrow{+0.99\%} 28.66\% \xrightarrow{+0.98\%} 29.64\%$ on GSM8K). This shows that the model actively evolves, refining its performance autonomously without additional manual intervention.

**(3) Persistent Efficacy of Self-Refinement:** Regardless of model variation, executing self-refinement consistently results in notable performance improvements. This shows that the self-refinement meta-capability learned by SELF is robust and consistent across various LLMs.

## 5 CONCLUSION

We present SELF (Self-Evolution with Language Feedback), a novel framework that enables LLMs to achieve progressive self-evolution through self-feedback and self-refinement. Unlike conventional methods, SELF transforms LLMs from passive information recipients to active participants in their evolution. Through meta-skill learning, SELF equips LLMs with the capability for self-feedback and self-refinement. This empowers the models to evolve their capabilities autonomously and align with human values, utilizing self-evolution training and online self-refinement. Experiments conducted on benchmarks underscore SELF's capacity to progressively enhance model capabilities while reducing the need for human intervention. SELF represents a significant step in the development of autonomous artificial intelligence, leading to a future in which models are capable of continual learning and self-evolution. This framework lays the groundwork for a more adaptive, self-conscious, responsive, and human-aligned future in AI development.

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

## A  APPENDIX

### A.1  IMPLEMENTATION DETAIL

#### A.1.1  TRAINING HYPERPARAMETERS

Our experiments were conducted in a computing environment equipped with 8 V100 GPUs, each having a memory capacity of 32GB. Below is a table 4 outlining the training hyperparameters we used. It is noted that these parameters were consistently applied across all training methods in our experiments.

Table 4: Training hyperparameters

| Hyperparameter | Global Batch Size | Learning Rate | Epochs | Max Length | Weight Decay |
|---|---|---|---|---|---|
| **Value** | 128 | $2 \times 10^{-5}$ | 3 | 2048 | 0 |

### A.1.2 DATA GENERATION

To produce the $D_{QA}$ dataset, we utilized 3.5k unlabeled training prompts for GSM8k and 2k training prompts [3] for the SVAMP. For the general test, we derived 6K conversations from a set of 90K ShareGPT dialogues to constitute the $D_{QA}$ data for the general test.

Regarding the prompts without labels used in the self-evolution training approach for math tests:

**First round self-evolving phase:** We made use of the leftover prompts from the training datasets, explicitly excluding those prompts that were utilized for meta-skill learning and labeled as $D_{QA}$. Specifically, we took 4K remaining prompts on GSM8k and 1K on SVAMP.

**Second/Third round:** We utilized the Self-Instruct method as described in (Wang et al., 2022b) We created unlabeled prompts using the template shown in Fig, A.1.2—initially, 4 to 6 instances served as seed examples. In the second round of self-evolution training, we produced 10K prompts, which was augmented to 15K in the third iteration.

In the general test, considering the need for the model to exhibit broad proficiency across various domains, we leveraged a subset (15K) of unlabeled prompts from ShareGPT dialogues to construct the self-evolution training data.

> You are an experienced instruction creator. You are asked to develop 3 diverse instructions according to the given examples.
> Here are the requirements:
> 1. The generated instructions should follow the task type in the given examples.
> 2. The language used for the generated instructions should be diverse.
> Given examples: {examples}
> The generated instructions should be:
> A. ...
> B. ...
> C. ...

### A.2 CASE STUDY ANALYSIS

This subsection delves into a detailed case study analysis that exhibits the comparative efficiencies of the original Vicuna and Vicuna + SELF models. As depicted in Fig. A.2, both models predict answers and engage in self-feedback and self-refinement processes. An interesting observation is that while Vicuna's refined answer retains inaccuracies, Vicuna + SELF generates a solution that's accurate and logically consistent by self-feedback and self-refinement.

### A.3 META-SKILL TRAINING CORPUS

As demonstrated in the provided case study within the A.3, the content represents a typical training example from our meta-skilling corpus. It showcases the initial response of the model, its self-feedback, and the subsequent refinement, enabling the model to self-feedback and self-refinement.

---

[3]Adhering to the official recommendation `https://github.com/arkilpatel/SVAMP/tree/main`, training prompts consist of MAWPS (Koncel-Kedziorski et al., 2016) and ASDiv-A (Miao et al., 2020)

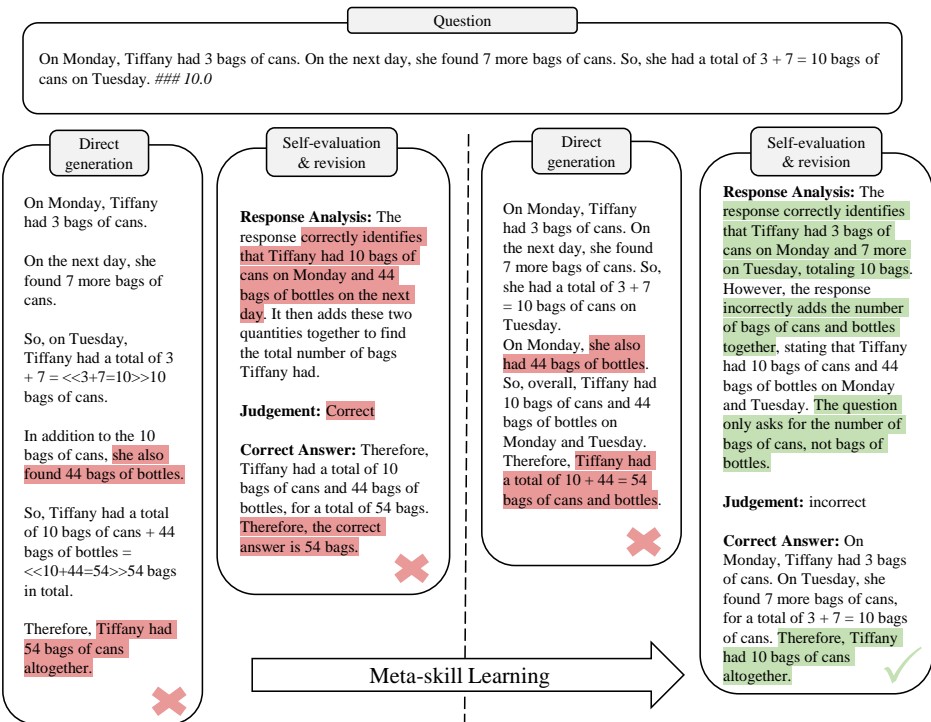

Figure 4: Case study of comparison between original Vicuna (left) and Vicuna+SELF (right) on an SVAMP. Both models generate direct predictions and undergo self-feedback and self-refinement. While Vicuna's refinement retains the incorrect answer, Vicuna+SELF showcases superior self-refinement capability, ultimately producing a correct and logically consistent solution.

**Question:** Greg has an alarm set to ring three times a day as a reminder. When the alarm goes off, it continues to ring until Greg turns it off. The first time it went off today, it rang four times. The second time it went off, it rang for three times as long as the first time. The third time, it rang for half as long as the second time. How many times did the alarm ring in all?
**Original Response:** The alarm rang four times the first time, three times as long as the first time the second time, and half as long as the second time the third time. So, the total number of times the alarm rang is $4 + 3*3 + 2*2 = 24$.
**Feedback:** The response correctly interprets the problem. The first time the alarm rang, it rang for 4 times. The second time it rang, it rang for 3 times as long as the first time, which is $3*4 = 12$ times. The third time it rang, it rang for half as long as the second time, which is $12/2 = 6$ times. However, the final calculation is incorrect. The total number of times the alarm rang is not $4 + 3*3 + 2*2 = 24$, but $4 + 12 + 6 = 22$. *Judgement: incorrect*
**Refined Response:** The alarm rang four times the first time, three times as long as the first time the second time, and half as long as the second time the third time. So, the total number of times the alarm rang is $4 + 12 + 6 = 22$.

## A.4 ALGORITHM

The subsequent algorithm, labeled as the "Two-Phase SELF Process", delineates a methodology to evolve a base language model using a progressively dual-phased approach: Meta-Skill Learning and Self-Evolving. Initially, the process involves training on a "Meta-Skill Learning corpus," which combines Question-Answer pairs and feedback-driven refinement data. After this phase, the algorithm proceeds to its "Self-Evolving Phase," where the model undergoes iterative refinements. The model employs data augmentation techniques for each iteration, generating self-refined outputs based on previously refined models. This self-evolving iteration is designed to capitalize on accumulated knowledge and refine the model using freshly generated data. The process culminates with an enhanced Language Model that has undergone multiple stages of self-evolution, showcasing im-

provements over its initial form. The detailed steps and mechanisms involved are delineated in Alg. 1.

---

**Algorithm 1:** Two-Phase SELF Process

---

**Data:** (1) Question-Answer pairs ($D_{QA}$), (2) Meta-Skill training data ($D_{meta}$) and (3) unlabeled prompts ($D_{unlabeled}$)
**Input**: An initial Language Model $M_{initial}$
**Result:** A stronger Language Model $M_{self}^k$ after self-evolving

`// Meta-Skill Learning Phase`
**Data:** Meta-Skill learning corpus ($D_{meta}$) and Question-Answer pairs ($D_{QA}$)
$M_{meta}$ = Supervised_fine_tuning($M_{initial}$, $D_{meta} \cup D_{QA}$ );

`// Self-Evolving Phase`
Initialize $M_1$ with $M_{meta}$;
**foreach** *iteration t in 1 to Number of self-evolving iterations T* **do**
    `// Data-Augmentation`
    Initialize $D_{\text{self}}^t$ as an empty set;
    **foreach** *prompt $p_{self}^i$ in $t^{th}$ Unlabeled prompts $D_{unlabeled}$* **do**
        Generate self-refined output $\hat{r}_{self}^i$ using $M_{self}^{t-1}$;
        Use $M_{self}^{t-1}$ to filter the self-refined output;
        Add $(p_{self}^i, \hat{r}_{self}^i)$ to $D_{\text{self}}^t$, where $r_i$ is the refined response;
    **end**
    $M_{self}^t$ = Supervised_fine_tuning($M_{self}^{t-1}$, $D_{self}^t$);
**end**

`// Training Complete`
**return** Improved Language Model $M_{self}^T$;

---

### A.5 PROMPT FOR GENERATING FEEDBACK AND REFINEMENT IN GENERAL CASE

For the general test, aligned with the methodology described in 3, we deploy the following prompt to guide an LLM-based annotator in generating response feedback and refinement. This prompt serves as the foundation for the meta-skill learning corpus and assists in producing self-evolution training data in the general test setting.

> **Prompt for feedback and refinement:**
> **(Feedback)** Please assess the quality of response to the given question.
> Here is the question: $p$.
> Here is the response: $r$.
> Firstly provide an analysis and verification for response starting with "Response Analysis:".
> Next, then rate the response on a scale of 1 to 10 (1 is worst, 10 is best) in the format of "Rating:"
> **(Refinement)** Finally output an improved answer based on your analysis if no response is rated 10.

### A.6 MULTIPLE V.S. SINGLE SELF-REFINEMENT

In this study, we examine the impact of two meta-skill training data organization methods on model performance: (1) Multiple Self-Refinement ($D_{FR-multi}$), which entails sampling three responses and instructing the model to select the best one for refinement, and (2) Single Self-Refinement ($D_{FR}$), where the model generates and refines only one response.

We present the comparative performance of these methods in Table 5. Our findings indicate that both methods benefit from an increased volume of training data, demonstrating performance improvements. Notably, as the data volume grows, the multiple-response refinement approach demonstrates a smaller improvement in direct generation performance (+4.02%) compared to the single-response method (+5.84%). Given the single-response method's simplicity and computational efficiency — requiring the sampling of only one response during inference — and its superior performance relative to the multiple-response approach, we have adopted the single-response refinement strategy in our experiments.

Table 5: Performance comparison between single and multiple response refinement across varying volumes of meta-skill training data. The right arrow indicates the performance improvement by self-refinement: "direct generation $\rightarrow$ self-refinement".

| Data Size | Vicuna + $D_{QA} \cup D_{FD}$ | Vicuna + $D_{QA} \cup D_{FD-multi}$ |
|---|---|---|
| 3.5k | $25.39 \rightarrow 28.28$ | $25.92 \rightarrow 27.29$ |
| 7.5k | $31.23 \rightarrow 32.98$ | $29.94 \rightarrow 32.14$ |

## A.7 SELF-EVOLUTION TRAINING DATA FILTERING ANALYSIS

Table 6: Analysis of filtering strategies on GSM8K. "Acc. of Training Data" refers to the accuracy of self-generated data post-filtering/refinement, while "Acc. on Test Set" indicates the model's test performance after fine-tuning such data.

| Filter Strategy | Acc. of Training Data (%) | Acc. on Test Set (%) |
|---|---|---|
| Self-Refinement Revised (Unfiltered) | 29.89 | 26.90 |
| Meta-Skill Filtered | **44.10** | **27.67** |

In Table 6, we explore the impact of different filtering strategies on the quality of training data and their contribution to self-evolution training. The following insights emerge from this comparison:

**(1) Superiority of Meta-Skills Filtered:** The combination of self-refinement and self-feedback filtering results in higher data accuracy (44.10%) and improved finetuned model performance (27.67%). Despite the significant accuracy boost, the performance gain is modest due to the reduced data size (from 4k to 1.8k) post-filtering.

**(2) Robustness of SELF:** The substantial accuracy increase in self-generated data with the addition of self-feedback meta-skill underlines its strong filtering capability, contributing to improved finetuned model performance.

## A.8 SELF-EVOLUTION TRAINING: CONTINUAL TRAINING V.S. RESTART TRAINING

Table 7: Analysis about varied self-evolution training methodologies on GSM8K

| Training Approach | Direct Generation (%) | Self-Refinement (%) |
|---|---|---|
| Base Model | 24.49 | 24.49 |
| Restart Training | **27.67** | **29.34** |
| Continual Training (Mixed Data) | 27.22 | 28.43 |
| Continual Training ($D_{self}^t$ Only) | 24.87 | 25.85 |

'Restart Training', which combines meta-skill learning corpus with all self-evolution training data, significantly improves direct generation (+3.18%) and self-refinement (+3.85%). This approach helps maintain a balance between new learning and previously acquired knowledge.

'Continual Training (Mixed Data)', where the model is trained simultaneously with self-evolution data from all rounds, also shows notable enhancements in direct generation (+2.73%) and self-refinement (+3.94%). In contrast, 'Continual Training ($D_{self}^t$ Only)', which trains the model sequentially with self-evolution data from each round, demonstrates more modest gains (+0.38% in direct generation, +0.98% in self-refinement). The relatively lower performance of the latter approach highlights the importance of a mixed data strategy for effective self-evolution training.

## A.9 SELF VS. SUPERVISED FINE-TUNING ON 7.5K GSM8K TRAINING DATA.

When fine-tuned on the GSM8K 7.5k training set, the Vicuna model achieves an accuracy of 35.70%, lower than SELF (37.87%).

Table 8: Comparison between SELF and Supervised Fine-Tuning

| Direct Generation (%) | Self-Refinement (%) | Meta-Skill Learning | | Self Evolution Process | |
|---|---|---|---|---|---|
| | | $D_{QA}$ | $D_{meta}$ | 1st round | 2nd round |
| 28.05 | - | ✓ | | | |
| 31.23 | 32.98 | ✓ | ✓ | | |
| 35.43 | 36.22 | ✓ | ✓ | ✓ | |
| **37.87** | **38.12** | ✓ | ✓ | ✓ | ✓ |
| 35.70 | - | SFT | (GSM8K training data) | | |

The result of 29.64% in Table 1 is derived from a meta-skill learning corpus of 3.5k. Experiments in Table 8 are conducted using an expanded 7.5k meta-skill data to ensure a fair comparison with the Supervised Fine-tuned model.

Table 8 shows that using 7.5k unlabeled training prompts to construct the meta-skill learning corpus, The baseline model Vicuna + $D_{QA}$ achieves 28.05%. After meta-skill learning, the result of direct generation is 31.23%, which improves to 32.98% after self-refinement. In subsequent self-evolution rounds, performance continues to improve, reaching 37.87% to 38.12% in the second round. This surpasses the result of supervised fine-tuning (35.70%).

**Continuous Improvement of SELF vs. Supervised Fine-tuning:** SELF's main advantage is its ability for continuous improvement and adaptation. Unlike supervised fine-tuning, SELF does not rely on human or external LLM (GPT3.5/GPT4) to annotate training data in the self-evolution training.

## A.10 SCALABILITY OF SELF FRAMEWORK

To explore how SELF performs with different starting model qualities, we conduct experiments using the OpenLlama-3b model (Geng & Liu, 2023), a smaller LLM along with a stronger LLM, VicunaV1.5(finetuned from Llama2-7b)l (Chiang et al., 2023), on the GSM8K dataset. This allows us to assess SELF's adaptability to model quality. Experiments with SELF are based on the first round of self-evolution. The results are as follows:

Table 9: Scalability of SELF Framework Across Different Models

| Model | Direct Generation (%) | Self-Refinement (%) |
|---|---|---|
| OpenLlama-3b | 2.04 | 1.01 |
| OpenLlama-3b + $D_{QA}$ | 12.13 | 10.97 |
| OpenLlama-3b + $D_{QA}$ + SELF | 15.32 | 15.78 |
| Vicuna (Llama-7b) | 16.43 | 15.63 |
| Vicuna + $D_{QA}$ | 24.49 | 24.44 |
| Vicuna + $D_{QA}$ + SELF | 27.67 | 29.34 |
| VicunaV1.5 (Llama2-7b) | 18.5 | 17.43 |
| VicunaV1.5 + $D_{QA}$ | 26.04 | 25.48 |
| VicunaV1.5 + $D_{QA}$ + SELF | **30.22** | **32.43** |

**Applicability and Robustness of SELF Framework:** The average improvement of 17.32% via direct generation and 16.87% after self-refinement underscores the framework's scalability and efficacy. It reveals a consistent positive impact of the SELF Framework across diverse models.

**SELF Framework exhibits enhanced performance on more powerful models:** In the table 9, applying SELF to VicunaV1.5 exhibits the most significant performance, 30.22% of direct generation and 32.43% of self-refinement compared to Vicuna and OpenLlama-3b. It is evident that as the underlying model's capabilities strengthen, the benefits introduced by the SELF framework also increase.

## A.11 IMPACT OF META-SKILL LEARNING QUALITY

We investigate how the quality of meta-skill learning influences the self-evolution process as follows:

Table 10: Comparison of Training Methods on GPT-3.5-turbo/GPT4

| Training Stage | Direct Generation (%) (GPT-3.5-turbo/GPT4) | Self-Refinement (%) (GPT-3.5-turbo/GPT4) |
|---|---|---|
| Vicuna + meta-skill learning | 24.84/25.39 (0.55↑) | 25.22/28.28 (3.06↑) |
| Vicuna + meta-skill learning + SELF | 25.11/27.67 (2.56↑) | 25.47/29.34 (3.87↑) |

The presented table 10 highlights substantial performance advancements achieved by employing GPT-4 to generate the meta-skill corpus within our SELF framework, as opposed to GPT-3.5-turbo. Specifically, the performance of direct generation and self-refinement exhibit noteworthy improvements across both training stages when utilizing GPT-4. For example, in the "Vicuna + meta-skill learning" phase, the result of direct generation increases from 24.84% (GPT-3.5-turbo) to 25.39% (GPT-4), reflecting a significant gain of 0.55%. Similarly, in the "Vicuna + meta-skill learning + SELF" stage, the result of self-refinement rises from 25.47% (GPT-3.5-turbo) to 29.34% (GPT-4), indicating a substantial enhancement of 3.87%.

This study underscores the crucial impact of high-quality meta-skill training data on Vicuna model performance within the SELF framework. Transitioning from GPT-3.5-turbo to GPT-4 for meta-skill corpus generation consistently improves Direct Generation and Self-Refinement metrics.

## A.12 SINGLE VS. MULTIPLE ROUNDS OF SELF-EVOLUTION

Given the same number of prompts, we compare the effect of training with a single round versus training iteratively, to assess the difference between a static and an improved model as a self-evolution training data generator as follows:

Table 11: Comparison of Single-Round Training and Iterative Training

| Training Method | Direct Generation (%) | Self-Refinement (%) |
|---|---|---|
| SELF (Single Round) | 28.40 | 30.55 |
| SELF (Iterative) | 29.64 | 31.31 |

Table 11 shows that in a single round, the performance is 28.40% for direct generation and 30.55% for self-refinement. The iterative approach shows higher scores (29.64%) for direct generation and 31.31% for self-refinement.

**Advantages of Iterative Training:** The iterative method benefits from improved LLMs in later rounds, producing higher-quality training data and, consequently, enhanced test performance.

