# OpenReview forum: "SELF: Language-Driven Self-Evolution for Large Language Model"
_ICLR.cc/2024/Conference — Submitted to ICLR 2024_

### Official Review · Reviewer_ARUW · 2023-10-18

**Soundness:** 3 good
**Presentation:** 1 poor
**Contribution:** 4 excellent
**Rating:** 5
**Confidence:** 2

**Summary:**

The authors propose a framework called SELF where an LLM is trained to acquire meta-skills that it applies to itself so as to improve its own performance on downstream tasks. The LLM is asked to refine its own output and learns from this refinement process, thus generating better and better output on various tasks, and still refining it. SELF has two processes (self-evolution, self-refinement), and the impact of these processes is studied in various datasets in comparison to and combination with a baseline called self-consistency, as well as with various ablations.

**Strengths:**

To me, the general approach is novel, quite original and potentially very fruitful. I'm convinced that, properly rewritten, this paper could have some impact.

The various comparisons, combinations and ablations studies efficiently shed light on the impact of the various processes, and provide a convincing picture of the approach. In particular I appreciate that the authors combined their approach with the self-consistency approach they compare theirs to.

If the authors manage to improve a lot the writing of the next version of their paper (see below), I'll be glad to change my rating towards acceptance.

**Weaknesses:**

The paper is poorly written at different levels and suffers from unclarities and from the lack of comparison with similar work. A tentative list:
- the only work the authors compare to is Wang et al. (2022a) about self-consistency, but this work is not even mentioned in the related work. It should be explained with some details.
- about related work, the authors ignore many attempts to use RL on large language models without human feedback, using the capability of LLMs to self-evaluate or some rewards coming from the task itself (see e.g. [1, 2] and [3] for some overview). A discussion of the difference to these related works and others found following the referenced papers in these works would be more than welcome.
- More experimental comparisons would make the paper stronger.
- there are many typos, some non-sentences, a lot of points are poorly written, I'll try to provide a list below, but the authors should find a way to improve a lot the way the paper is written, either using grammar tools or the help of stronger scientific writers. In particular, I think that the introduction could put more forward the many messages that can be extracted from the empirical study.


- [1] Pang, J. C., Wang, P., Li, K., Chen, X. H., Xu, J., Zhang, Z., & Yu, Y. (2023). Language Model Self-improvement by Reinforcement Learning Contemplation. arXiv preprint arXiv:2305.14483.
- [2] Carta, T., Romac, C., Wolf, T., Lamprier, S., Sigaud, O., & Oudeyer, P. Y. (2023). Grounding large language models in interactive environments with online reinforcement learning. arXiv preprint arXiv:2302.02662.
- [3] Sun, H. (2023). Reinforcement Learning in the Era of LLMs: What is Essential? What is needed? An RL Perspective on RLHF, Prompting, and Beyond. arXiv preprint arXiv:2310.06147.

**Questions:**

- can the authors explain how the approach of Wang et al. (2022a) works and how it is related to their work?

- how does the SELF method relate to methods applying reinforcement learning without human feedback to improve LLMs? Could some of these methods be compared experimentally of the same datasets?

- "SELF facilitates the acquisition of self-refinement ability in smaller LLMs": this sentence is confusing in several respects. Do you mean that SELF can only be applied to small LLMs? Or that the SELF framework can be used in a context where the refiner network improves another, smaller downstream network? In both cases, this is raising questions that the paper does not answer to: (1) would the method work with larger LLMs? If SELF can be applied in a context where the refiner network improves another, smaller downstream network, what are the corresponding experimental results?

- Related work, about RLHF. I don't understand the sentence "RLHF involves complicated iteration between the models and reward functions, requiring many hyper-parameters tuning". Can you explain better what you mean? Provide a reference?  Another problem that the authors do not put forward is the availability of humans to perform RLHF.

- Are LLMs so good at self-feedback? We often read that many of them are very certain to be correct when in fact they are completely wrong. Can you back up the claim that they are good at self-feedback with references? Won't there be many counter-examples?

- How does your method prevent overfitting? When do you stop training and self-refining?

- How many examples in the EvolInstruct testset? You said it for all other datasets.

- Could you explain Fig. 3? What do the colors mean, what should we see, how was it obtained?

- In 4.4: "We present a comparison between utilizing the entire self-curated data—Unfiltered (4k)—and employing self-filtered data" -> what is the difference between self-curated and self-filtered? I don't understand the point here...

# Local issues and typos: #

I would add an "s" at the end of the title (models).

these models' innate potential: these models are not biological systems, can we say that they have some innate potential? Don' you mean "intrinsic"?

As depicted in Fig 2 and Fig 1 -> reverse order

refinement. thereby (remove dot)

Section 3.1 starts with "We observe the base Vicuna" but nothing has been said about Vicuna before, this sentence comes out of the blue.

It's -> It is

The beginning of Section 3.2 and 3.2.1 is full of typos:
- the model progressively self-evolving -> evolves
- the model M_meta generate and refine -> generates and refines
- for the evolve iteration t -> evolution
- with each instance in this augmented corpus is noted -> remove "is"
- self-evolution, We initialize -> we

Are shown in 3. -> Figure ? Table ? Section ?...

"For an in-depth understanding of each column’s meaning and significance." -> this is not a sentence, something is missing

less evident for ”Continual Training (D^t_self)” -> ”Continual Training (D^t_self Only)”

---

> ### Author Response · Authors · 2023-11-17
> **Response to Reviewer ARUW (1/4)**
>
> **W1 & Q1: Discussion about self-consistency**
>
> Self-consistency [[1]](#1)  works by sampling a diverse set of reasoning paths and then selecting the most consistent answer (We sample 5 times in our experiments), a straightforward and effective method. Self-refinement enables models to evaluate and improve their outputs.
>
> Self-consistency operates without additional finetuning while self-refinement applies to broader domains, not just those with unique correct answers, as demonstrated in Section 4.2.2 (General Test).
>
> Section 4.2.1. shows that self-refinement can complement self-consistency.
> Integrating these two strategies leads to a better performance.
>
> We will incorporate a thorough discussion and citation of [[1]](#1) in the revised paper.
>
> [1] Xuezhi Wang, Jason Wei, Dale Schuurmans, Quoc Le, Ed Chi, Sharan Narang, Aakanksha Chowdh-
> er, and Denny Zhou. Self-consistency improves chain of thought reasoning in language models.
> arXiv preprint arXiv:2203.11171, 2022a.
>
>
> **W2: Missing reference of RL-without human feedback**
>
> We will revise our paper to include a discussion on RL-related methods in the 'Related Works' section.
> We have conducted an additional RL experiment, the results of which will be detailed and compared in the response to Q2.

---

> ### Author Response · Authors · 2023-11-17
> **Response to Reviewer ARUW (2/4)**
>
> **W3: More comparison experiments**
>
> We have conducted 6 additional experimental comparisons to form a more solid work.
> We will include details of these experiments in our revised paper.
> The key experiments include:
>
> - RL-based Baseline Comparison: We compare the SELF framework with RL-based baselines (Response to Q2).
>
> - Self-Consistency Filtering Analysis: This explores the contribution of meta-skills and self-consistency to the generation of the self-evolution training corpus.
>
>     | Model | Acc. of Training Data(\%) | Acc. on Test set(\%) |
>     |--------------|----------|----------|
>     |  self-consistency filtered (5x majority)   | 28.27            | 26.77 |
>     | self-refinement revised     | 29.89                 | 26.90 |
>     | meta-skills filtered         | **44.10**                |  **27.67** |
>     ||
>
>     This study demonstrates self-refinement as a necessary component of SELF. Meta-skills comprising self-refinement and self-feedback form a robust foundation for the self-evolution training process.
>
> - SELF Adaptability with OpenLlama3b/VicunaV1.5(Llama2-7b): This examines the adaptability and extensibility of SELF across models of varying sizes and capacities  (Response to Q.3).
>
> - Impact of Meta-Skill Learning Quality: We investigate how the quality of meta-skill learning influences the self-evolution process.
>     | Training Stage   | Direct Generation(%, GPT-3.5-turbo/GPT4) | Self-Refinement(%, GPT-3.5-turbo/GPT4)  |
>     |-------------------------------------------|--------------------|-----------------------------------|
>     | Vicuna + meta-skill learning       |  24.84/25.39 (0.55$\uparrow$)    |  25.22/28.28 (3.06$\uparrow$)        |
>     | Vicuna + meta-skill learning + SELF |  25.11/27.67 (2.56$\uparrow$)         |  25.47/29.34 (3.87$\uparrow$)               |
>     |                                           |
>
>     The experiments showcase better performance metrics across our SELF framework by using GPT-4 to generate the meta-skill corpus, compared with GPT-3.5-turbo.
>     This study affirms the critical role of meta-skill training data quality in instilling self-feedback and self-refinement in the Vicuna model.
>
> - Single vs. Multiple Rounds of Self-Evolution: Given the same number of prompts, compare the effect of training with a single round versus training iteratively, to assess the difference between a static and an improved model as a self-evolution training data generator.
>     | Training Method                 | Direct Generation (\%) | Self-Refinement (%) |
>     |---------------------------------|------------------------|-----------------|
>     | SELF (Single Round)                     | 28.40                 | 30.55          |
>     |  SELF (Iterative)         | 29.64                 | 31.31         |
>
>
>     The comparison reveals a higher performance of iterative training over that of single round training.
>     It highlights the advantages of iterative training in leveraging improved LLMs across rounds for enhanced training data quality and subsequent test performance.
>
> - SELF with 7.5k Meta-Skill Corpus vs. Supervised Fine-Tuning on 7.5k GSM8k training data.
>
>     | Training Method                 | Direct Generation(\%) | Self-Refinement(%) |
>     |---------------------------------|------------------------|-----------------|
>     | Vicuna + $D_{QA}$(7.5k)      |  28.05      | - |
>     | Vicuna + meta-skill learning (7.5k)      | 31.23         | 32.98 |
>     | Vicuna + meta-skill learning (7.5k)  + first round  Self-Evolution  |   35.43                | 36.22          |
>     | Vicuna + meta-skill learning (7.5k)  + first and second round  Self-Evolution  |   **37.87**               | **38.12**         |
>     |||||
>     | Vicuna  + GSM8K training data (human-annotated, 7.5k) | 35.70 | - |
>     ||
>
>     In comparison to supervised fine-tuning on the GSM8K 7.5k training set, the SELF approach achieves a higher accuracy of 37.87%, surpassing the fine-tuned model's accuracy of 35.70%. It's essential to highlight that the reported 29.64% in Table 2 of our paper stemmed from a 3.5k meta-skill learning corpus and to ensure a fair comparison, new experiments using an expanded 7.5k meta-skill data demonstrate the effectiveness of SELF, with performance reaching 38.12%, outperforming the supervised fine-tuning result.
>
> These experimental results will be detailed in the appendix of the revised paper due to the scope limitation.

---

> ### Author Response · Authors · 2023-11-17
> **Response to Reviewer ARUW (3/4)**
>
> **W4: Poorly written & Local issues and typos:**
>
> We appreciate your guidance regarding the local issues and typos and the suggestions for incorporating more empirical study findings into the Introduction of our paper.
> We will thoroughly review each suggestion and make revisions.
> We aim to release the revised paper as soon as possible.
>
> Moreover, we would like to clarify your concerns regarding the last point in **Local issues and typos**:
>
>     'Restart Training' involves combining the meta-skill learning corpus with all rounds of self-evolution training data to train the model.
>
>     "Continual Training (Mixed Data)" involves training the model with self-evolution data from all rounds.
>
>     'Continual Training ($D^t_{self}$ Only).' refers to training the model with the $t$th round self-evolution data in a sequential manner, round by round.
>
>     We will update the manuscript to reflect this distinction more clearly
>
>
> **Q2: Comparison with reinforcement learning based method**
>
> SELF differs from reinforcement learning (RL) methods in its use of informative **natural language feedback** instead of information-sparse **scalar rewards**.
>
> We also conducted an additional RL-based experiment based on [trlx](https://github.com/CarperAI/trlx). For a fair comparison, we use the same SFT model (Vicuna + $D_{QA}$) for SELF and RLHF. The reward model was trained on pair-wise comparison data (refined response is assumed better than original response) derived from the meta-skill learning corpus in SELF.
> Although we follow the RLHF framework to conduct experiments,
> the comparison data was provided by GPT4 instead of humans, which can be seen as a form of reinforcement learning without human feedback.
> The results are as follows:
>
>
>
>
> | Method | GSM8K_test(%) |
> |--------|----------------------|
> | SFT | 24.49 |
> | RL   | 25.55                |
> | SELF   | **27.67**            |
>
>
> **RL vs. SELF**:  RL achieved a 25.55\% accuracy on GSM8K, lower than SELF's 27.67\%.
> We found that the reward model often fails to identify the correct response, leading to limited performance improvements. For instance, **76\%** of incorrect answers were assigned higher scalar rewards than correct answers on the GSM8K test set.
> Unlike RL-based methods, SELF leverages natural language feedback, which provides a more accurate evaluation (only **28\%** incorrect answers were judged as correct).
>
>
> **Q3: Apply SELF to stronger LLMs and the variant of SELF**
>
> Previously, it was widely believed that self-improvement capabilities were exclusive to large language models (LLMs) such as GPT-4 and GPT-3.5. However, our research in the SELF framework reveals that even smaller models can acquire such abilities.
> This is evidenced in Table 1, which shows that initial Vicuna models lacked self-improvement capabilities, yet our meta-skill training effectively endowed it with self-refinement abilities.
>
>
> We did not mean SELF can only be used in small LLMS.
> Instead, we have observed improvement of SELF in a more robust baseline model, i.e., VicunaV1.5, which was fine-tuned from Llama2-7b.
>
> To address your question, we experiment with SELF with OpenLlama-3b(smaller LLM), VicunaV1.5(stronger LLm), and Vicuna(in paper), demonstrating its effectiveness across different models.
>
> | Model | Direct Generation(\%) | Self-Refinement(%) |
> |---------------------------------|------------------------|-----------------|
> | OpenLlama-3b | 2.04  | 1.01 |
> | OpenLlama-3b + $D_{QA}$ |   12.13  | 10.97 |
> | OpenLlama-3b + $D_{QA}$ +  SELF (one round self-evolution)|  15.32    | 15.78 |
> ||||
> | Vicuna (Llama-7b)  | 16.43 | 15.63 |
> | Vicuna + $D_{QA}$ | 24.49 | 24.44 |
> | Vicuna + $D_{QA}$ +  SELF (one round self-evolution)| 27.67 | 29.34 |
> ||||
> | VicunaV1.5 (Llama2-7b) | 18.5  | 17.43 |
> | VicunaV1.5 + $D_{QA}$ | 26.04 | 25.48 |
> | VicunaV1.5 + $D_{QA}$ +  SELF (one round self-evolution)| **30.22**  |  **32.43**   |
> ||
>
> The method of "a refiner network improves another, smaller downstream network" can seen as a variant of SELF. It is interesting to explore in future work.
>
>
> **Q4: Confusion about RLHF's limitation and one more limitation of RLHF(limited human availability)**
>
> "RLHF involves complicated iteration between the models and reward functions, requiring many hyper-parameters tuning\" means: challenges in reinforcement learning, particularly the iterative inference between policy and reward models and the complexity of numerous hyperparameters (e.g., minibatches, discount factor, etc.).
>
> We will update the 'Related Work' section of our paper to provide references and a clearer description of these aspects and also discuss the expensive human availability of RLHF.

---

> ### Author Response · Authors · 2023-11-17
> **Response to Reviewer ARUW (4/4)**
>
> **Q5: Are LLMs good at self-feedback**
>
> To support the claim that LLMs are capable of self-feedback, we can refer to studies like [[1](#1),[2](#2),[3](#3)]. These references suggest that LLMs can indeed engage in meaningful self-improvement.
>
> we found that meta-skill learning is critical to the self-feedback ability.
> After meta-skill learning, the accuracy of self-feedback to identify the correctness of an answer is **70\%** (Without meta-skill learning, the accuracy is limited to **33\%**, as the model tends to self-feedback all responses as correct.).
>
> We presume that self-feedback ability might depend on several factors, such as the model's intrinsic ability, prompt design, problem domain, etc.
>
>
> [1] William Saunders, Catherine Yeh, Jeff Wu, Steven Bills, Long Ouyang, Jonathan Ward, and Jan Leike. Self-critiquing models for assisting human evaluators. arXiv preprint arXiv:2206.05802, 2022.
>
> [2] Noah Shinn, Federico Cassano, Beck Labash, Ashwin Gopinath, Karthik Narasimhan, and Shunyu
> Yao. Reflexion: Language agents with verbal reinforcement learning, 2023.
>
> [3] Aman Madaan, Niket Tandon, Prakhar Gupta, Skyler Hallinan, Luyu Gao, Sarah Wiegreffe, Uri Alon, Nouha Dziri, Shrimai Prabhumoye, Yiming Yang, et al. Self-refine: Iterative refinement
> with self-feedback. arXiv preprint arXiv:2303.17651, 2023.
>
>
>
>
> **Q6: When to stop training and self-refinement**
>
> we use common regularization techniques to prevent overfitting, including monitoring validation set performance and early stopping.
>
> Regarding self-refining during inference,  we consistently apply self-refinement once across all models for a fair comparison.
> Besides, our additional tests indicate optimal performance with 3-4 self-refinement iterations.
> Beyond 4 iterations, performance may decline.
>
>
> **Q7: Test cases in EvolInstruct test set**
>
> The EvolInstruct test set comprises 218 test examples, We will include this information in section 4.1.1 of the revised paper.
>
>
> **Q8: Figure 3 explaination**
>
> Figure 3 shows the comparison test result in the general domain. As noted in sec. 4.2.2, "We follow the evaluation procedures outlined in [[1]](#1), which address the order bias issues identified in the evaluation methods proposed by [[2]](#2)". Specifically, we test different models (Vicuna, Vicuna fine-tuned on $D_{QA}$, apply SELF to Vicuna fine-tuned on $D_{QA}$, referred to as Vicuna, Vicuna + $D_{QA}$, and Vicuna + $D_{QA}$ + SELF) on the Vicuna test set and EvolInstruct test set.
> We then employed GPT-4 to assign an assessment score for each test response of different models.
>
> In Figure 3, the color coding is as follows:
>
> - Blue indicates test cases where the model under evaluation performs better than the baseline model (Vicuna), as judged by GPT-4.
>
> - Yellow indicates test cases of equal performance.
>
> - Pink indicates test cases where where model under evaluation performs worse than the baseline model.
>
> From this figure, several key findings emerge:
>
> - First Block: After being fine-tuned on $D_{QA}$, the model demonstrates improved performance compared to the original Vicuna model, as evaluated by GPT-4.
>
> - Second Block: The application of the SELF framework to Vicuna + $D_{QA}$ enhances performance on both test sets.
>
> - Third Block: Further improvements are observed when Vicuna + $D_{QA}$ + SELF applied self-refinement during inference.
>
> These results demonstrate that SELF is effective in the general domain.
> We recognize the need for clarity and will add more detailed explanations in our paper to facilitate better understanding.
>
> [1] Can Xu, Qingfeng Sun, Kai Zheng, Xiubo Geng, Pu Zhao, Jiazhan Feng, Chongyang Tao, and Daxin Jiang. Wizardlm: Empowering large language models to follow complex instructions. arXiv preprint arXiv:2304.12244, 2023.
>
> [2] Wei-Lin Chiang, Zhuohan Li, Zi Lin, Ying Sheng, Zhanghao Wu, Hao Zhang, Lianmin Zheng, Siyuan Zhuang, Yonghao Zhuang, Joseph E. Gonzalez, Ion Stoica, and Eric P. Xing. Vicuna: An open-source chatbot impressing gpt-4 with 90%* chatgpt quality, March 2023. URL https://lmsys.org/blog/2023-03-30-vicuna/
>
> **Q9: Confusion of 'self-curated ' and 'self-filtered'**
>
> Apologies for the confusion. Here's a clarification:
>
> **self-curated data**:
> This is the data after applying self-refinement to self-generated responses.
>
> **self-filtered data**: This data undergoes one extra self-feedback step to the self-refinement response.
> We filter out the data that is judged by self-feedback as incorrect.
>
> **The purpose of Section 4.4**: We are investigating whether self-feedback can further improve the self-evolution training data quality and thus result in better test performance.
>
> We will update the paper to better explain these concepts and our experiment settings.

---

> > ### Comment · Reviewer_ARUW · 2023-11-18
> > **Please upload a revised version asap**
> >
> > My concerns being a lot about the way the paper is writtent, I won't change my perspective about this paper before reading a new version. ICLR provides a great opportunity to upload a new version and discuss it further with reviewers, the authors should make profit of it.

---

> > > ### Comment · Reviewer_ARUW · 2023-11-19
> > > **Remarks on new version**
> > >
> > > My general feeling is that the paper has improved, but the writing is still not good enough, so I keep my score so far. Here are a few examples of issues below. If the authors take seriously the need  for another approach to improve the writing, they may improve the paper far beyond the few examples given below.
> > >
> > > Besides, a good idea would have been to highlight the changes with colors to facilitate re-reading.
> > >
> > > Poor writing:
> > >
> > > In contrast, our proposed SELF diverges in its objective. -> the "in contrast" and "diverges" are redundant, and the sentence does not say much.
> > >
> > > RLHF involves complicated iteration between the models and reward functions, -> one complicated iteration? Can you explain in what sense it is "complicated"?
> > >
> > > [...]
> > >
> > > a novel framework signifying a promising advancement in the autonomous self-evolution of LLM development. -> poor wording.
> > >
> > >
> > > Typos:
> > >
> > > Again, the authors should use tools or the help of expert authors to improve the writing. Some of the typos outlined below would be found by standard grammatical tools.
> > >
> > > Fascinatingly, recent study (Ye et al., 2023) -> a recent study
> > >
> > > there are research efforts propose to improve LLMs output quality via Online Self-improvement -> research efforts WHICH propose...
> > >
> > > Meanwhile, Sun (2023) discuss the integration of conventional RL with LLMs. -> discusses (if there is only one author). BTW, you should avoid using a paper as subject in a sentence, it is better to put a ref at the end on the sentence whenever possible.

---

> > > > ### Author Response · Authors · 2023-11-21
> > > > **Further Response to Reviewer ARUW**
> > > >
> > > > Thank you for your detailed response and suggestions.
> > > > We greatly appreciate your detailed guidance on enhancing our manuscript's writing during the review process.
> > > > We have addressed the issues you pointed out and extensively improved the writing throughout the manuscript.
> > > > In our newly uploaded paper, we have highlighted **major** changes in blue to facilitate re-reading and conserve your review time.
> > > >
> > > > If you find any remaining issues in our writing, please do not hesitate to point them out. We are very willing to continue making revisions.

---

### Official Review · Reviewer_9itr · 2023-10-31

**Soundness:** 2 fair
**Presentation:** 1 poor
**Contribution:** 2 fair
**Rating:** 3
**Confidence:** 5

**Summary:**

The authors introduce "SELF," which enables the continual enhancement of the abilities of LLMs. In this approach, the model first learns two meta-skills: self-feedback and self-refinement. Afterwards, the model can autonomously generate responses from given unlabeled instructions. Moreover, the model can be trained further on these instructions and filtered responses to achieve better improvement. Additionally, during inference, self-refinement can be utilized to attain better performance. Experiments conducted on GSM8K, SVAMP, Vicuna, and Evol-Instruct demonstrate the effectiveness of their method.

**Strengths:**

1. The idea of the proposed pipeline is sound.
2. The experiments in the paper demonstrate the effectiveness of SELF.

**Weaknesses:**

1. The paper lacks organization. Some essential details are absent, making it difficult to reproduce the results. Below are some sample questions that need to be addressed in the paper:
     a) What is the detailed pipeline for collecting the training corpus for meta-skill learning (self-feedback, self-refinement)?
     b) Which model is employed to generate feedback and refine answers produced by $M_{initial}$?
     c) What are the hyperparameters used during $M_{meta}$'s training?

2. The second and third rounds of self-evolution utilize self-instruct to generate new questions, whereas the first round only employs the original questions. Is there a particular reason for this discrepancy? Furthermore, if we were to combine all this data into a single round, would the performance be comparable to that of multiple rounds?

3. Table 2 indicates that, during the meta-skill learning stage, the model undergoes training over QA data with ground-truth labels, significantly enhancing the QA performance. Yet, section 3.1 mentions that meta-skill learning doesn't encompass training over QA. How does the paper reconcile this inconsistency?

4. The second row of Table 2 reveals that training over $D_{meta}$ can also boost reasoning performance. Is this improvement attributed to the use of self-refinement during inference? If so, what would the performance be without utilizing self-refinement during inference?

5. The author should compare SELF to a supervised fine-tuning baseline. What is the performance of fine-tuning Vicuna over the GSM8K training set? Will its performance surpass that of SELF? According to the paper titled "Scaling Relationship on Learning Mathematical Reasoning with Large Language Models," direct fine-tuning of the model llama-7b over the GSM8K training set appears to achieve approximately 35% accuracy. This figure is better than the results depicted in Table 2. What advantages does the intricate "SELF" pipeline offer in practice? A potentially simpler alternative might involve gathering quality responses with more robust models like GPT-3.5/GPT-4 and fine-tuning the model over those results.

In summary, while the proposed pipeline is interesting, the authors need to provide additional materials to support their claims and outcomes.

**Questions:**

Here are the revised statements:

1. Is self-refinement a necessary component for SELF? Table 1 indicates that SELF with self-consistency is already sufficient. In fact, self-refinement can sometimes even degrade performance. To fully justify the necessity of self-refinement, it is recommended to add an ablation study on its use in self-evolution part.
2. Some related work on using LLMs for debugging/checking is absent.
    - Teaching Large Language Models to Self-Debug
    - Deductive Verification of Chain-of-Thought Reasoning.

---

> ### Author Response · Authors · 2023-11-17
> **Response to Reviewer 9itr (1/2)**
>
> **W1(a): Training corpus collection pipeline**
>
> In Section 3.1.1 of our paper, we have outlined the pipeline for collecting the training corpus for meta-skill learning, including self-feedback and self-refinement.
> In the revised version of our paper, we will provide more comprehensive details.
>
> **W1(b): Model for generating feedback and refinement**
>
> For generating the meta-skill learning corpus, we employed GPT-4 owing to its superior refinement capabilities
>
> **W1(c): Hyperparameters**
>
> The hyperparameters used during the training were described in Appendix A.1.1 of our paper.
> These parameters were consistently applied across all training methods in our experiments.
>
>
> **W2: Combining all data into a single round versus multiple rounds (iterative)**
>
> To address your concerns, we conducted the following experiments:
>
>
> | Training Method                 | Direct Generation (\%) | Self-Refinement (%) |
> |---------------------------------|------------------------|-----------------|
> | SELF (Single Round)                     | 28.40                 | 30.55          |
> |  SELF (Iterative)         | 29.64                 | 31.31         |
>
> **Single Round vs. Iterative Training**: In a single round, the performance is 28.40% for direct generation and 30.55% for self-refinement.
> The iterative approach shows higher scores: 29.64% for direct generation, 31.31% for self-refinement.
>
> **Advantages of Iterative Training**: The iterative method benefits from improved LLMs in later rounds, producing higher-quality training data and, consequently, enhanced test performance.
>
> **W3: About training over QA data during the meta-skill learning stage**
>
> (1) The QA data in Table 2, denoted as $D_{QA}$, consists of pairs of prompts and pseudo answers provided by GPT4.
> It does not include ground-truth labels, which are noted in section 3.1.
> Moreover, pseudo answers in $D_{QA}$ are extracted from meta-skill learning data $D_{meta}$.
>
>
> (2) We combine both $D_{QA}$ and $D_{meta}$ as we explain in **Section 3.1**: "Given that the data structure in $D_{meta}$ diverges from conventional direct question answering formats, we also employ a dataset composed of pairs of questions and refined answers, denoted as  $D_{QA}$, during meta-skill training."
>
> To eliminate any confusion, we will revise our paper, specifically Section 3.1, to clearly explain how $D_{QA}$ and $D_{meta}$ contribute to the meta-skill learning.
>
> **W4: $M_{\text{meta}}$ can also boost reasoning performance**
>
> In the caption of Table 2, "The right arrow indicates the performance improvement by Self-Refinement" indicates that the table displays results both **without** and **with** the application of self-refinement.
> The second row shows the result '25.39 → 28.28'.
> Here, the 25.39% reflects the **direct generation** performance, which is an improvement without self-refinement compared with the 24.49% (first row).
> The subsequent increase (+2.89%) to 28.28% represents the additional gain brought by self-refinement.
> It is evident that the inclusion of the meta-skill learning corpus ($D_{\text{meta}}$) leads to enhancements in both direct generation and self-refinement outcomes.
>
> As we have discussed in Section 4.3(1) ("Integration of Meta-skill Training Data $D_{\text{meta}}$ Elevates Direct QA"), providing language feedback to the mistake of the model can improve the performance.
> The positive impact of $D_{\text{meta}}$ is an interesting finding and is worthy of future research.

---

> ### Author Response · Authors · 2023-11-17
> **Response to Reviewer 9itr (2/2)**
>
> **W5: SELF versus Supervised Fine-tuning baseline**
>
> In addressing the comparison between the SELF framework and supervised fine-tuning using the GSM8K training set, we present the following results:
>
> | Training Method                 | Direct Generation(\%) | Self-Refinement(%) |
> |---------------------------------|------------------------|-----------------|
> | Vicuna + $D_{QA}$(7.5k)      |  28.05      | - |
> | Vicuna + meta-skill learning (7.5k)      | 31.23         | 32.98 |
> | Vicuna + meta-skill learning (7.5k)  + first round  Self-Evolution  |   35.43                | 36.22          |
> | Vicuna + meta-skill learning (7.5k)  + first and second round  Self-Evolution  |   **37.87**               | **38.12**         |
> |||||
> | Vicuna  + GSM8K training data (human-annotated, 7.5k) | 35.70 | - |
> ||
>
> **Supervised Fine-Tuning vs SELF**:  The Vicuna model, when fine-tuned on the GSM8K 7.5k training set achieves an accuracy of 35.70%, which is lower than of SELF  (37.87%).
>
> It is important to note that the 29.64% in Table 2 of our paper originated from a 3.5k meta-skill learning corpus. To ensure a fair comparison with the Supervised Fine-tuned model, new experiments were conducted using an expanded 7.5k meta-skill data as shown above.
>
> Specifically, using 7.5k unlabeled training prompts to construct the meta-skill learning corpus,
> The baseline model Vicuna + $D_{QA}$ achieves 28.05%.
> After meta-skill learning, the initial result is 31.23%, which improves to 32.98% after self-refinement. The performance further increases in subsequent self-evolution rounds, with the second round reaching 37.87% to 38.12%, surpassing the supervised fine-tuning result(35.7%).
>
> **Advantages of SELF**:
> SELF's main advantage is its ability for continuous improvement and adaptation.
> Unlike supervised fine-tuning, SELF does not rely on human or external LLM (GPT3.5/GPT4) to annotate training data in the self-evolution training.
> As noted in the response to weakness 4, providing language feedback to the model's incorrect responses can also improve its question-answering (QA) capabilities.
> This is another advantage of SELF.
>
> **Q1: Self-refinement is a necessary component for SELF**
>
> We note that performance degradation occurs only in models lacking meta-skill learning in Table 1 of our paper (Vicuna, and Vicuna with $D_{QA}$).
> In contrast, models equipped with meta-skill learning consistently exhibit performance improvement via self-refinement.
> Table 1 demonstrates that SELF, when applied with self-refinement, achieves an accuracy of 31.31%, surpassing the 29.87% obtained using self-consistency.
> Furthermore, while self-consistency is limited to problems with unique answers, self-refinement can be applied to a more general domain as evidenced in Section 4.2.2(General Test).
>
> We also conducted an ablation study to compare the effect of self-refinement and self-consistency on self-evolution training data quality and their subsequent impact on test performance:
>
> | Model | Acc. of Training Data(\%) | Acc. on Test set(\%) |
> |--------------|----------|----------|
> |  Self-Consistency Filtered (5x Majority)   | 28.27            | 26.77 |
> | Self-Refinement Revised     | 29.89                 | 26.90 |
> | Meta-Skills Filtered         | **44.10**                |  **27.67** |
> ||||
>
> In the above table, "Acc. of Training Data" refers to the accuracy of self-generated training data post-filtering/refinement, while "Acc. on Test Set" indicates the model's test performance after fine-tuning such training data.
>
> As shown in the table, self-refinement produces higher quality training data compared with self-consistency, and results in better fine-tuned model performance.
> The final row demonstrates that further filtering the self-refinement data with self-feedback can improve both training data accuracy and test performance.
>
> The study clearly shows that self-refinement is a **necessary component** of SELF.
> Coupled with self-feedback, self-refinement establishes a robust foundation for the self-evolution training process.
> Moreover, self-refinement can improve the model's performance during inference.
>
> **Q2: Some related work for debugging/checking is missing**
>
> We appreciate your suggestion.
> We will update our paper to add citations of these works into related works and discuss how SELF is related.

---

> > ### Comment · Reviewer_9itr · 2023-11-20
> > **Thanks for the response!**
> >
> > Thank you for the experiments and explanations provided. Despite these efforts, I feel that my concerns haven't been entirely addressed, so I will not change my initial score. Here are some comments.
> > > 1. The structure of the paper is still not good. It's difficult to capture all the crucial details when reading the paper.
> > > 2. The paper includes a lot of experiments and numbers, which is good. However, explanations are not clear, resulting in some confusion.

---

> > > ### Author Response · Authors · 2023-11-21
> > > **Further Response to Reviewer 9itr**
> > >
> > > Thank you very much for taking the time to review our work. We have uploaded a new version of the paper, in which we have made efforts to improve our writing and presentation. The major changes have been highlighted in blue to facilitate re-reading.
> > >
> > > We hope you can take a moment to review the modifications we have made with effort. We would be grateful if you could kindly provide a clear description of any remaining issues in our writing, as we are eager to continue making revisions.

---

> ### Comment · Reviewer_9itr · 2023-11-22
> **Final response**
>
> Thank you for your efforts! However, I must admit that after spending a considerable amount of time reading the paper, I am still quite confused. I couldn't grasp the core contribution of the paper initially. I suggest that the authors rewrite **sections 3 & 4** and resubmit to the next conference. **The current version cannot be accepted by ICLR.**

---

### Official Review · Reviewer_VDXd · 2023-11-01

**Soundness:** 4 excellent
**Presentation:** 3 good
**Contribution:** 3 good
**Rating:** 6
**Confidence:** 4

**Summary:**

This paper proposes a novel framework called SELF (Self-Evolution with Language Feedback) to enable large language models (LLMs) to self-evolve and improve their capabilities over time. In detail, the paper proposes to 1) equip LLMs with meta-skills for self-feedback and self-refinement through meta-skill learning. This allows models to evaluate their own outputs and refine them; 2) Use the meta-skills to generate high-quality training data via self-curated responses and iterative refinement; 3) Conduct self-evolution training where models iteratively fine-tune on self-curated data to enhance capabilities; 4) Apply online self-refinement during inference to further improve response quality. Experiments on math and general domain benchmarks demonstrate SELF can consistently improve model performance through self-evolution cycles. The learned meta-skills also enable smaller models to acquire advanced self-refinement abilities.

**Strengths:**

1. The idea of empowering LLMs with meta-skills for autonomous self-improvement is interesting.
2. The iterative process of self-generated data, training, and online refinement is intuitive and aligns well with human learning.
3. Results verify SELF consistently improves performance over baseline models, and that meta-skills boost self-refinement capability.

**Weaknesses:**

After rebuttal: Many thanks for the rebuttal. I decide to keep my original score. Nevertheless, there are a few minor questions below that I would encourage you to address in your next revision.
- I just noticed that this idea of "self-evolution" seems to be similar to "self-training" [1][2], so I'd encourage you to briefly describe the differences.
- Also, how many rounds did you use? How to determine the number of self-evolution rounds, by experiment? What are the criteria for ending self-evolution?
- Why do you use three rounds? What are the results of rounds 4, 5, and 6?
- Finally, does using self-evolution significantly increase training and inference time, and what is the training/inference time?

[1] Self-training with Noisy Student improves ImageNet classification. 	CVPR 2020.

[2] Revisiting Self-Training for Neural Sequence Generation. ICLR 2020

Original reviews:
1. The quality of meta-skills relies on the initial annotator model/human. No analysis on sensitivity to this factor.
2. Limited insight on how self-evolving training affects model internals and learned representations.
3. More comparisons to related human preference alignment methods would be useful.

**Questions:**

1. How robust is SELF to noise in self-generated data? Are the meta-skills strong enough to filter bad data?
2. Is there an upper limit or plateau to the self-evolution process? How to tell when to stop?
3. For real-world deployment, how to prevent unsafe or unethical knowledge from entering self-evolving training?
4. How dependent is SELF on starting model quality? Could it work for simple baseline models?
5. How does the computational overhead of SELF compare to regular supervised training? Is it more expensive?

---

> ### Author Response · Authors · 2023-11-17
> **Response to Reviewer VDXd (1/2)**
>
> **W1: The analysis on sensitivity of SELF to the quality of meta-skills.**
>
> Acknowledging the importance of the quality of meta-skill training data, we conducted experiments to examine the effect of using different models (GPT-3.5-turbo vs GPT-4) for corpus construction.
>
> | Training Stage   | Direct Generation(%, GPT-3.5-turbo/GPT4) | Self-Refinement(%, GPT-3.5-turbo/GPT4)  |
> |-------------------------------------------|--------------------|-----------------------------------|
> | Vicuna + meta-skill learning       |  24.84/25.39 (0.55$\uparrow$)    |  25.22/28.28 (3.06$\uparrow$)        |
> | Vicuna + meta-skill learning + SELF |  25.11/27.67 (2.56$\uparrow$)         |  25.47/29.34 (3.87$\uparrow$)               |
> |                                           |
>
> The experiments indicate that with a higher quality meta-skill learning corpus from GPT4 compared with GPT-3.5-turbo, all performance metrics in our SELF framework show improvement. This is particularly significant when applying self-refinement, where the increase in higher quality meta-skill data surpasses 3%.
>
> Our results confirm that the quality of the meta-skill training data is crucial for instilling self-feedback and self-refinement into the Vicuna model.
> These findings underscore the importance of using high-quality models for corpus construction in meta-skill training.
>
> **W2: Insight on how self-evolving training affects model internals and learned representations.**
>
> We have conducted thorough experiments to investigate how different factors influence self-evolving training.
> Exploring how self-evolving training affects model internals and learned representations is a worthwhile research direction.
> Due to limited resources and scope, we leave this for future work.
>
> **W3: Comparisons to related human preference alignment methods.**
>
> We have conducted additional experiments to compare the SELF framework with Reinforcement Learning from Human Feedback (RLHF).  For a fair comparison, we use the same SFT model (Vicuna + $D_{QA}$) for SELF and RLHF.
> The reward model was trained on pair-wise comparison data (refined response is assumed better than original response) derived from $D_{meta}$, the meta-skill learning corpus in SELF.
> The results are as follows:
>
> | Method | GSM8K_test(%) |
> |--------|----------------------|
> | SFT | 24.49 |
> | RLHF   | 25.55                |
> | SELF   | **27.67**            |
>
> The experimental analysis is as follows:
>
> RLHF achieved a 25.55\% accuracy on GSM8K, lower than SELF's 27.67\%.
> The reason is that the reward model often fails to identify the correct response via a single scalar value, leading to limited performance improvements. Specifically, 76\% of incorrect answers were assigned higher scalar rewards than correct answers on the GSM8K test set.
> Unlike RLHF, SELF leverages natural language feedback, which provides a more accurate evaluation (only 28\% incorrect answers were judged as correct).
>
> **Q1: The robustness of SELF to noise in self-generated data and the effectiveness of meta-skills in filtering.**
>
> We conduct the following experiments to illustrate the robustness of SELF to noise in self-generated data and the effectiveness of meta-skills in filtering:
>
> | Model | Accuracy of Training Data(\%) | Accuracy on Test set(\%) |
> |--------------|----------|----------|
> | Unfiltered    | 29.89      | 26.90 |
> | Meta-Skill Filtered         | **44.10**                           |  **27.67** |
>
> The second column means the accuracy of self-generated training data.
> The third column means the test performance of the model finetuned on such data.
>
> Experimental analysis:
>
> (1) Unfiltered: This strategy utilizes self-refinement to refine the self-generated data without filtering.
>
> (2)  Meta-Skills Filtered: After applying self-refinement, we utilize self-feedback to filter the data, and there is a significant improvement in the accuracy of training data (44.10\%). The performance of the finetuned model also improves (27.67\%).
> We observe that the performance improvement is not as significant as the accuracy improvement of self-generated data.
> We hypothesize the reason is that the self-generated data size shrinks after applying filtering (from 4k to 1.8k).
>
> By solely using self-refinement, the performance of the finetuned models still improves.
> This shows the robustness of SELF to noise in self-generated data.
> The accuracy of the self-refinement data significantly improves with the help of self-feedback meta-skill and results in better fine-tuned model performance. This shows that self-feedback has a strong ability to filter bad data.

---

> ### Author Response · Authors · 2023-11-17
> **Response to Reviewer VDXd (2/2)**
>
> **Q2: An upper limit or plateau in the self-evolution process and when to stop**
>
> Table 1 in our paper illustrates that iteratively enhancing model capabilities with increased self-evolution training corpus progressively improves performance. However, we presume that this improvement may exhibit diminishing returns over time.
>
> To determine the optimal point for halting the self-evolution, we employ a validation set to monitor performance variance. When the model performance converges, indicating minimal gains from additional training, it is appropriate to stop the self-evolution process.
>
> Additionally, the upper bound of self-evolution effectiveness is influenced by several factors. We have done additional experiments that indicate that stronger baseline models lead to more significant improvements in self-evolution performance, thereby raising their learning upper limits. The result is as follows:
>
> | Model                              | Initial GSM8k Result | Meta-Skill Learning(direction generation ->self-refinement) | Self-Evolution(direction generation ->self-refinement)|
> |------------------------------------|----------------------|---------------------|----------------|
> | VicunaV1.5 (finetuned from Llama2) | 18.5                 | 27.19 -> 29.27      | 32.63 -> 34.20 |
> | Vicuna(shown in paper)             | 16.43                | 25.39 -> 28.28      | 29.64 -> 31.31 |
>
> This indicates that starting with a more robust model can enhance the efficacy of the self-evolution process.
>
> **Q3: Prevent the inclusion of unsafe or unethical knowledge in the self-evolving training**
>
> To prevent unsafe or unethical knowledge from entering self-evolving training, we can train the model with meta-skills to recognize and revise unethical or unsafe content.
> We then apply these skills during the self-evolution phase.
> By integrating this mechanism into the self-evolution process, we aim to ensure that the model remains aligned with ethical and safe guidelines.
>
> **Q4: Dependency of SELF on the starting model quality**
>
> To explore how SELF performs with different starting model qualities, we conducted experiments using the OpenLlama-3b model [[1]](#1), a smaller LLM, along with a stronger LLM, VicunaV1.5(finetuned from Llama2-7b), on the GSM8K dataset. This allowed us to assess SELF’s adaptability to model quality. The results were as follows:
>
> | Model | Direct Generation(\%) | Self-Refinement(%) |
> |---------------------------------|------------------------|-----------------|
> | OpenLlama-3b | 2.04  | 1.01 |
> | OpenLlama-3b + $D_{QA}$ |   12.13  | 10.97 |
> | OpenLlama-3b + $D_{QA}$ +  SELF (one round self-evolution)|  15.32    | 15.78 |
> ||||
> | Vicuna (Llama-7b)  | 16.43 | 15.63 |
> | Vicuna + $D_{QA}$ | 24.49 | 24.44 |
> | Vicuna + $D_{QA}$ +  SELF (one round self-evolution)| 27.67 | 29.34 |
> ||||
> | VicunaV1.5 (Llama2-7b) | 18.5  | 17.43 |
> | VicunaV1.5 + $D_{QA}$ | 26.04 | 25.48 |
> | VicunaV1.5 + $D_{QA}$ +  SELF (one round self-evolution)| **30.22**  |  **32.43**   |
> ||
>
> We conclude that SELF demonstrates robustness by consistently enhancing the performance of models from smaller to stronger.
> Its efficacy is affected by the capability of the base model, as the stronger model acquires more benefits from SELF.
>
> [1] Xinyang Geng and Hao Liu. Openllama: An open reproduction of llama, May 2023. URL https://github.com/openlm-research/open_llama
>
>
> **Q5: Computational overhead and more expensive cost of SELF compared to regular supervised training**
>
>  The SELF framework consists of two primary steps:
>
> 1) Creating self-evolution training data.
>
>     In contrast to regular supervised fine-tuning, which often requires extensive manual data annotation—a laborious and time-consuming process—SELF automates the generation of its training data. This automation significantly reduces the effort and time typically associated with data annotation.
>
>
> 2) Fine-tuning the model with this newly generated data.
>
>     It's important to note that the actual training phase in SELF is similar to the Supervised Fine-Tuning (SFT) process. Therefore, SELF does not introduce additional computational costs in the training phase.
>
> When considering the overall process, SELF effectively reduces both the time and monetary costs associated with data collection, making it a more efficient alternative to traditional supervised training methods.

---

### Author Response · Authors · 2023-11-19
**To All Reviewers**

We would like to express our sincere gratitude to all the reviewers for your valuable time, insightful suggestions, and constructive comments. We are truly appreciative that ALL reviewers provide positive remarks about our work, such as **"The idea of empowering LLMs with meta-skills for autonomous self-improvement is interesting."** (Reviewer VDXd), **"The idea of the proposed pipeline is sound and experiments in the paper demonstrate the effectiveness of SELF"** (Reviewer 9itr), and **"The general approach is novel, quite original, and potentially very fruitful. This paper could have some impact if properly written."** (Reviewer ARUW).

We have addressed each of the reviewers' comments in a detailed manner. Please let us know in time if you are satisfied with our response.

We have also uploaded a new version of our paper. A comprehensive list of the main changes is provided below for your convenience:

In the main paper:

- 1 Introduction: We have expanded the discussion of our empirical findings in the penultimate paragraph.
- 2 Related Works: We added more discussions on related work and missing references (Self-Consistency, RL-based methods, and self-debugging/checking) and reorganized the section.
- 3 Method: We improved our writing and provided more comprehensive details, e.g., a detailed pipeline for collecting the training corpus for meta-skill learning. We also revised inappropriate wording, incomplete sentences, and typos for better readability.
- 4 Experiments:
   - We enhanced our writing and presentation.
   - We added RL-based Baseline Comparison with SELF in section 4.2.2, as all reviewers were concerned about this issue.
   - We relocated some experiments, along with newly conducted ones, to the Appendix due to space limitations (Note: we did not remove any experiments from the original paper).
   - If you have previous questions about any experiment, you can locate the corresponding experiment by result number or by following the outline at the beginning of Section 4.
   - We have detailed each term for a better understanding, e.g., a detailed description of the general test.

In the appendix:

- We added important supplementary experiments to better verify the SELF framework:
  1) Effect of different meta-skill training corpus construction methods.
  2) Impact of utilizing filtering strategies when constructing the self-evolving corpus.
  3) Impact of divergent self-evolution training strategies (restart training vs. continual training).
  4) SELF vs. Supervised Fine-tuning.
  5) Scalability of SELF with different starting model qualities.
  6) Influence of the quality of meta-skill learning data on self-evolution training.
  7) Effect of training in a single round of self-evolution versus training iteratively.

Best regards,

Paper 662 Authors

---

### Meta-Review · Area_Chair_qBDV · 2023-12-03

**Metareview:**

The paper introduces SELF, a framework that enables LLMs to self-evolve and improve their capabilities over time. SELF accomplishes this by equipping LLMs with the meta-skills of self-feedback and self-refinement, allowing them to self-curate data and refine their own responses. Through iterative self-evolution cycles of generating responses, self-evaluation, and fine-tuning, LLMs progressively improve their performance. Although the paper certainly has its merit, the reviewers generally have concerns about the writing quality. I have read the revised version to confirm this point. Although there has been improvements from the initial version, it seems obvious that there is room to improve for paper structure, description clarity, and gammer (there are actually quite some in the revised text). In addition, I think that the key question from Reviewer VDXd regarding the limitation of the improvement process has not been fully addressed. I encourage the authors to incorporate more qualitative and quantitative analysis in future versions of this paper. I'd lean to rejecting the current submission.

**Justification For Why Not Higher Score:**

Writing quality has certain room for improvement. Insufficient analysis.

**Justification For Why Not Lower Score:**

N/A

---

### Decision · Program_Chairs · 2024-01-16

Reject